# ONE TOKENIZER TO RULE THEM ALL: EMERGENT LANGUAGE PLASTICITY VIA MULTILINGUAL TOKENIZERS

## ABSTRACT

Pretraining massively multilingual Large Language Models (LLMs) for many languages at once is challenging due to limited model capacity, scarce high-quality data, and compute constraints. Moreover, the lack of language coverage of the tokenizer makes it harder to address the gap for new languages purely at the post-training stage. In this work, we study what relatively cheap interventions early on in training improve "language plasticity", or adaptation capabilities of the model post-training to new languages. We focus on tokenizer design and propose using a *universal* tokenizer that is trained for more languages than the primary pretraining languages to enable efficient adaptation in expanding language coverage after pretraining. Our systematic experiments across diverse groups of languages and different training strategies show that a universal tokenizer enables significantly higher language adaptation, with up to 20.2% increase in win rates compared to tokenizers specific to pretraining languages. Furthermore, a universal tokenizer also leads to better plasticity towards languages that are completely unseen in the tokenizer and pretraining, by up to 5% win rate gain. We achieve this adaptation to an expanded set of languages with minimal compromise in performance on the majority of languages included in pre-training.

## 1 INTRODUCTION

There are only a handful of research labs with enough compute resources and expertise to train large AI systems at scale (Maslej et al., 2025; Hooker, 2024). Most researchers and practitioners are forced to choose among available pretrained models for downstream tasks, even if they are not tailored to their use cases. Nowhere is this tension more evident than in the multilingual setting (Joshi et al., 2020; Singh et al., 2024; Üstün et al., 2024), where limited investment in multilingual support in pretraining often results in significant gaps in language coverage in state-of-the-art LLMs (Holtermann et al., 2024).

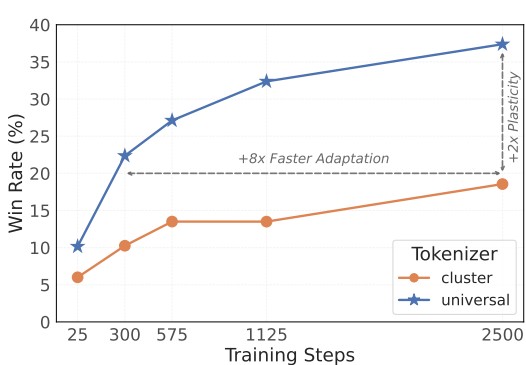

Figure 1: UNIVERSAL tokenizer exhibits **+2x higher plasticity** with **+8x faster adaptation**.

This imbalance in language coverage has created a growing divide in the cost of use for particular language users as marginalized languages require more tokens and incur higher latency for generations (Ji et al., 2023; Cui et al., 2024; Ahia et al., 2023), restricting speakers of low-performing languages to lower quality technology (Held et al., 2023; Durmus et al., 2024; Nicholas & Bhatia, 2023; Ojo et al., 2025). Further compounding these issues, once a model is pretrained, it is hard to steer towards new behavior using post-training alone (Wang et al., 2025). Unless the tokenizer has been calibrated to a new language during training, it often requires far more significant amount of data and intricate optimization steps (Muller et al., 2021).

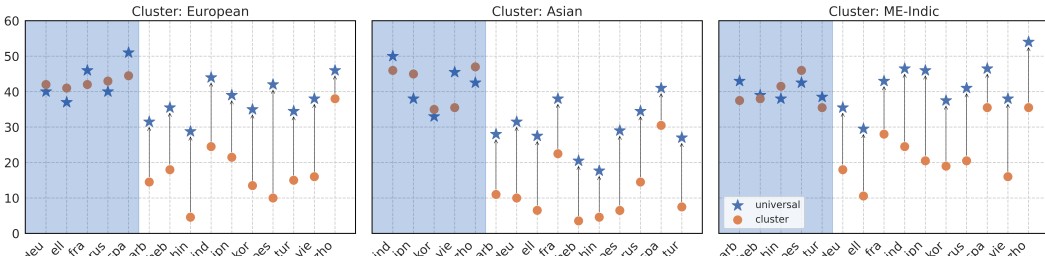

Figure 2: Win rates for models trained with the UNIVERSAL and CLUSTER tokenizers against Dolly generations (Singh et al., 2024). The shaded blue portion represents a subset of primary languages, and the white represents expanded languages.

**Multilingual plasticity** represents the capability of the language model to quickly adapt to lingual distribution shifts to the downstream target, which in our case, involves a new set of focus languages (Chen et al., 2023). Given that pretraining requires the bulk of compute and cost resources, any intervention made at this stage that improves the *plasticity* for downstream developers and researchers is beneficial.

In this work, we investigate minimal and efficient pretraining interventions to reduce later adaptation costs. In particular, we identify tokenization as an area with relatively low cost of intervention, but potential for large downstream gains. We ask: *Can we leverage tokenizers with broad language coverage to improve the plasticity of LLMs without hurting pretraining performance?*

We hypothesize that a universal tokenizer that is trained on more languages than the primary pretraining languages, introduced from the start of pretraining, enables quick and effective interventions for adapting a model to new languages. This significantly diverges from the previous work that focuses on techniques such as vocabulary extension (Wang et al., 2020) or retraining the embedding layer (Artetxe et al., 2020) after pretraining. These techniques are more costly, needing more resources like training budget, with varying degrees of success across languages (Limisiewicz et al., 2023; Sharthak et al., 2025; Nag et al., 2025).

We systematically investigate the impact of multilingual tokenizers through an exhaustive set of ablations at pretraining scale, which requires a significant investment of resources. We vary tokenizer, language subsets, and adaptation strategies across 69 languages, and we find the following results:

1. The **UNIVERSAL** tokenizer significantly improves adaptation to new (expanded) languages, achieving an average of 19% higher win rate in continued pretraining experiments, compared to the baseline tokenizers specialized to pretraining languages. In addition to achieving higher adaptation, the **UNIVERSAL** tokenizer exhibits almost the same performance on primary languages with no more than 2% difference in downstream evaluation against the baseline tokenizer.

2. For targeted adaptation where the new languages are the only focus, the **UNIVERSAL** tokenizer achieves an average of 14.6% improvement over the baseline tokenizer for the expanded languages subset. Furthermore, for adapting to fully **unseen** languages, not included both in the tokenizer and pretraining (the most extreme case of adaptation), UNIVERSAL tokenizer outperforms the baseline by up to 5% gain in win rates across 7 heavily under-resourced languages.

3. We find that the UNIVERSAL tokenizer enables more than 8x faster adaptation performance, requiring much less additional training, and therefore minimal costs. We believe that this dramatically benefits practitioners who want to extend the language coverage of a pretraining model with minimal intervention.

## 2 METHODOLOGY AND EXPERIMENTAL SETUP

### 2.1 METHODOLOGY AND CORE ABLATIONS

**Language Coverage and Model Variants.** Our experiments include 62 typologically and lexicographically diverse languages, broken up into three geographically motivated clusters: (1) European

languages, (2) Asian languages, and (3) Middle-Eastern and Indic languages (referred to as ME-Indic throughout the paper). For each geo-cluster, we pretrain language models primarily on the languages within that cluster (referred to as **primary** subset) and we use the remaining languages (referred to as **expanded** subset) as reference points for plasticity adaptation experiments. For example, for the European cluster, the primary subset consists of languages such as Spanish, Russian, and Portuguese, and the expanded subset includes the 10 languages outside of the dominant training data for that cluster. In addition, we also consider 7 **fully unseen** languages, such as Sinhala and Kazakh, which were not present in the tokenizer or base model training data. The full language list with clusters is provided in Appendix H.

**Adaptation Strategies.** One of our goals is to introduce highly plastic and adaptable model properties. A practitioner may choose to conduct language adaptation using a variety of different training strategies, influenced by what data they may have available. Ideally, the choices we make for the tokenizer allow for improved plasticity given any approach taken after pretraining. Hence, we evaluate our interventions under various different adaptation strategies, which include continued pretraining with data in both **primary and expanded** language subsets, targeted adaptation for **expanded** languages, and targeted adaptation for **fully unseen** languages. We briefly describe both these strategies and experimental details below:

- **Continued pretraining with data from primary and expanded languages:** The objective for this strategy is to increase language coverage of the model, so that it supports both primary and expanded languages. Half of the training mix consists of an even distribution of all languages in the instruction finetuning data, and the other half is a standard cooldown mix with high-quality datasets (See § 2.2). This imparts instruction-following abilities to our base models, and also allows for evaluation on both the primary and expanded languages.

- **Targeted adaptation (expanded languages):** In these experiments, we explore a targeted language adaptation through supervised fine-tuning. The post-training data solely consists of instruction-style data in the expanded language subsets for each cluster model. This allows us to isolate the effect of introducing new languages that are not focused on during pretraining but represented in the tokenizer.

- **Targeted adaptation (fully unseen languages):** In the final set of experiments, we explore the most extreme setting of targeted adaptation to fully unseen languages that are not seen in the tokenizer or pretraining. In this setting, we consider the availability of the data in one language only for each experiment, thus, we fine-tune the base model on one language at a time. This ablation enables evaluating our approach for adaptation under a heavily under-resourced scenario.

**Tokenizer Variants.** We train a massively multilingual tokenizer using data from all 62 languages as well as cluster-specific tokenizers that represent only the primary language subsets. Throughout the paper, we refer to these tokenizers as UNIVERSAL and CLUSTER tokenizers, respectively. We include more details about the tokenizer training in Section 2.3.

## 2.2 EXPERIMENTAL SET-UP

**Pretraining datasets.** Models are pretrained with a mixture of English, code, and multilingual corpora, where data weights are distributed as 55%, 15%, and 30%, respectively. Upweighting English in multilingual training is a common practice, due to higher task coverage and quality, which is crucial for cross-lingual transfer (Dash et al., 2025; Singh et al., 2024; Ravisankar et al., 2025). We also include code data as it has become a standard part of the training recipe even for natural language models, and has been found to boost performance on other tasks when included in pretraining (MA et al.; Aryabumi et al., 2024). We use a large corpus of data from a variety of public and proprietary sources. For models we pretrain with the UNIVERSAL tokenizer, we reallocate 5% of the training mixture from English data and uniformly distribute it among all the expanded languages, to avoid undertrained tokens in the vocabulary (Land & Bartolo, 2024). However, in Section 5.4, we ablate this percentage and show that even when no expanded language subset data is included in pretraining, the UNIVERSAL tokenizer significantly improves multilingual plasticity.

**Cooldown and instruct datasets.** For continued pretraining, we use cooldown data that involves upweighting higher quality datasets, comprised of text, math, code, and instruct-style data (Aryabumi et al., 2024). It has been found by recent work to improve performance on downstream tasks, in

particular by helping impart instruction-following capabilities (Parmar et al., 2024; Team et al., 2025). We include a high-quality mix of proprietary and open data, much of which was created by following a multilingual data arbitrage strategy (Odumakinde et al., 2024), covering 100,000 prompt-completion pairs in 23 languages. Finally, for experiments on fully unseen languages, we emulate a realistic data-constrained regime, and use only 14,800 instructions per language from the translated Dolly training set from Aya Collection (Singh et al., 2024).

**Training details.** For our experiments, as standard for most LLMs, we use the Transformer-based decoder-only architecture (Vaswani et al., 2017; Radford & Narasimhan, 2018). Our architecture includes key optimizations such as Parallel Attention Blocks (Chowdhery et al., 2023), Grouped Query Attention (Ainslie et al., 2023), SwiGLU activation function (Shazeer, 2020), and Rotary Positional Embeddings (Su et al., 2024). Additional training and infrastructure details are provided in Appendix B.

## 2.3 Tokenizer Training

All tokenizers are trained using the Byte Pair Encoding algorithm (Sennrich et al., 2016). Additional implementation details about the tokenizer training is given in Appendix A. We use a vocabulary size of 250k tokens in our main experiments, although we experiment with various sizes in § 5.3.

**Language weighting.** In addition to varying the coverage of the tokenizer, with some being trained on UNIVERSAL and CLUSTER language coverage, we also invest in a methodology that adjusts the weighting based upon availability of data. In contrast to traditional approaches which sample uniformly across all data and end up dominated by most frequent languages, we consider two factors: (1) natural distribution of

$$w_i = \frac{w_i^d . w_i^b}{\sum_n w_n^d . w_n^b} \quad (1)$$

the data available across languages, and (2) language buckets formed by languages that share the same family and script (which are more likely to share tokens). Within each language bucket, we use uniform weighting across languages. Concretely, for a language $i$, where $w_i^d$ and $w_i^b$ denote weights for data distribution and language bucket, respectively, we compute the language weights in the tokenizer data mixture following equation 1.

This way, we balance natural data distribution (skewed through the high-resource languages) with language bucketing in a principled manner, ensuring that there is equitable representation for diverse scripts and lower-resourced languages. Our pretraining experiments (Section 3, Appendix E) show that our specialized weighting combining language bucketing with size-proportional data distribution enables better compression ratios than uniform weighting and achieves better downstream performance. For the remainder of this work, we use specialized weighting throughout experiments, unless specified otherwise.

## 2.4 Evaluation

**Open-ended evaluation.** Goldman et al. (2024) find that generative tasks are more informative than classification in evaluating tokenizers, likely due to the number of generation steps. Following Üstün et al. (2024), the quality of generations is assessed using LLM-as-a-Judge win rates, where original generations are used as the reference answer. We use the `dolly_human_edited` and the `dolly_machine_translated` splits of the Aya Evaluation Dataset (Singh et al., 2024) as test data for this task, which are formed by translating 200 held-out examples from the Dolly-15k (Conover et al., 2023). We use 15 adaptation languages for open-ended evaluation (Appendix H).

Prior work has shown that LLMs as evaluators are reasonable proxies and aligned with human preferences also in multilingual settings (Üstün et al., 2024; Singh et al., 2025; Dang et al., 2024; Kreutzer et al., 2025). We use Command-A (Cohere et al., 2025) as the judge model, given its reported strength as the best open-weights judge on multilingual setting, scoring closely to GPT4o (Gureja et al., 2024; Pombal et al., 2025). The full judge prompt is included in Appendix D.3.

**Task-specific performance.** We use two task specific evaluations for multilingual evaluations. Belebele (Bandarkar et al., 2024) is a multiple-choice question machine-reading comprehension (MRC) dataset representing 122 language variants. Multilingual MMLU (M-MMLU) (Dac Lai et al., 2023) is a machine-translated version of the original MMLU dataset (Hendrycks et al., 2021) that contains questions ranging in topic from STEM to humanities.

## 3 RESULTS ON PRETRAINING PERFORMANCE

In this section, we first benchmark the performance of our pretrained models, to ensure that using a UNIVERSAL tokenizer doesn't cause degradations in performance, as may be expected when using a tokenizer optimized for a broader set of languages than the primary set.

**UNIVERSAL tokenizer does not compromise performance on primary languages.** As seen in Table 1, we find that our expanded UNIVERSAL tokenizer is remarkably competitive against CLUSTER across the geo-cluster models. The difference in pretraining performance is less than at most 1% average accuracy in Eng-

| Cluster | Tokenizer | Belebele | M-MMLU | EN Tasks |
|---------|-----------|----------|--------|----------|
| | | **PRIMARY LANGUAGES** | | |
| European | CLUSTER | 41.4 | 31.1 | 48.5 |
| | UNIVERSAL | 41.9 | 30.9 | 48.4 |
| Asian | CLUSTER | 38.2 | 29.6 | 48.2 |
| | UNIVERSAL | 38.1 | 28.9 | 48.1 |
| Middle East & Indic | CLUSTER | 38.1 | 29.2 | 49.1 |
| | UNIVERSAL | 36.5 | 28.6 | 48.2 |

Table 1: Performance of pretrained models using the CLUSTER vs. UNIVERSAL tokenizers.

lish tasks. The highest performance difference between UNIVERSAL and CLUSTER tokenizers for multilingual tasks is only 1.6% average accuracy on Belebele for ME-Indic cluster (38.1% vs 36.5 %). Overall, we observe minimal trade-offs in performance on primary cluster languages switching to UNIVERSAL tokenizer.

In fact, we observe that the UNIVERSAL tokenizer leads to a slight increase on average on Belebele for Euro cluster (41.9 vs 41.4) and achieves much closer performance for Asian cluster (38.1 vs 38.2). As additional validation, Figure 6 in Appendix D.1 shows the progression of average Belebele performance for both tokenizers for Euro cluster models during pretraining. The UNIVERSAL tokenizer achieves approximately similar performance throughout the whole pretraining, also suggesting the same trend in a longer pretraining run. Overall, these results showcase that using a UNIVERSAL tokenizer doesn't spell any significant performance degradation in pretraining for the primary languages.

**Balanced language weighting with language buckets for tokenizer training leads to better pretraining performance.** As described in Section 2.3 on tokenizer training, we weight the languages using buckets formed by script and language family, balanced against data availability. In order to motivate this weighting scheme, we compare pretraining performance of UNIVERSAL tokenizer against a baseline tokenizer (UNIFORM), where all languages are uniformly weighted except English. We conduct this ablation in the Euro cluster, where the number of primary languages is the highest. In tokenizer training, we use all the languages (62 languages; primary and expanded subsets) and only vary the language weighting. As shown in Table 3, UNIVERSAL tokenizer with balanced weighting using language buckets outperforms UNIFORM weighting in 21 European languages out of 27, with a relative gain of 2.2% (41.9 vs 41.0) on average. Further validating pretraining results, we provide the comparison for compression performance between these two tokenizers in Appendix E, where the results show better overall compression in UNIVERSAL tokenizer.

## 4 RESULTS ON ENHANCED MULTILINGUAL PLASTICITY

### 4.1 BENEFITS OF PLASTICITY IN CONTINUED PRETRAINING

In this section, we ask: *Does varying the approach for the tokenizer lead to plasticity benefits after continued pretraining on both primary and expanded languages?*

**Models trained with the UNIVERSAL tokenizer demonstrate significantly higher win rates on the EXPANDED SUBSET.** Figure 2 and Table 2a show results of evaluation across the Euro, Asian, and ME-Indic clusters, with 5 languages belonging to each PRIMARY LANGUAGE SUBSET and 10 in the EXPANDED SUBSET belonging to the other two clusters. We see that the UNIVERSAL tokenizer achieves an average gain of 18.9% in win rates across all three geo-cluster models on the expanded subsets, compared to models trained with the CLUSTER tokenizer. Improvement is consistent across clusters, where we find +19.9%, +17.8%, and +18.9% increase in win rate for Euro, Asian, and ME-Indic cluster models, respectively. Among all the expanded languages, Persian (+25.8%), Hindi (+23.3%), and Vietnamese (+22.0%) show the highest benefit from the UNIVERSAL tokenizer in the Euro, Asian, and ME-Indic clusters, respectively.

**(a) Continued pretraining**

| Cluster | Tokenizer | Dolly Win Rates (%) PRIMARY | EXPANDED | Flores-200 (BLEU) PRIMARY | EXPANDED |
|---|---|---|---|---|---|
| European | CLUSTER | 42.8 | 17.6 | 23.5 | 8.7 |
| | UNIVERSAL | 42.8 | 37.4 (+19.9) | 24.3 | 13.4 (+4.7) |
| Asian | CLUSTER | 41.7 | 11.7 | 11.3 | 7.7 |
| | UNIVERSAL | 41.8 | 29.5 (+17.8) | 14.2 | 15.7 (+8.0) |
| Middle East & Indic | CLUSTER | 39.7 | 22.8 | 12.3 | 13.2 |
| | UNIVERSAL | 40.2 | 41.8 (+18.9) | 15.1 | 16.5 (+3.3) |

**(b) Targeted adaptation**

| | Dolly Win Rates (%) CLUSTER | UNIVERSAL |
|---|---|---|
| European | 27.2 | 37.4 (+10.2) |
| Asian | 18.8 | 34.3 (+15.7) |
| Middle East & Indic | 23.31 | 41.1 (+17.8) |

Table 2: (a) The UNIVERSAL tokenizer matches CLUSTER performance on primary languages and shows large gains (up to 19.9% winrates in Dolly and up to 8 BLEU score in Flores) on average of expanded language subsets across all clusters. (b) Win rates on expanded languages after **targeted adaptation**. The UNIVERSAL tokenizer shows better performance (up to 17.8%) across all clusters over the baseline CLUSTER tokenizer.

**UNIVERSAL tokenizer preserves performance on all geo-clusters.** While the UNIVERSAL tokenizer provides significant gains on the expanded languages, in Figure 2, we observe that the performance on primary languages is nearly the same across both tokenizers in all three clusters. There is only a 0.3% win rate difference across all clusters for primary languages when comparing tokenizers, where UNIVERSAL tokenizer even leads to a slight increase over the CLUSTER tokenizer in the European, Asian, and ME-Indic models by 0.3%, 0.1%, and 0.5%, respectively. This is beneficial, as it suggests no trade-offs of improving plasticity for an expanded set of languages by using the UNIVERSAL tokenizer for the primary languages a provider is interested in when developing a model.

**In machine translation, models trained with the UNIVERSAL tokenizer outperform CLUSTER baseline across all the langauges.** To extend evaluation beyond LLM as a judge, we evaluate our cluster models trained with UNIVERSAL and CLUSTER tokenizers after continued pretraining on Flores-200 (Team et al., 2022), for both primary and extended languages. We find that the advantage of the UNIVERSAL tokenizer is exhibited in this evaluation as well, with 5.3 average increase in BLEU score on the expanded languages, and even an improvement in the primary languages of 2.2 BLEU.

## 4.2 BENEFITS OF PLASTICITY IN TARGETED ADAPTATION

An experimental setting of great interest is the more realistic scenario where a downstream developer only has access to data in the EXPANDED languages. To mimic this scenario, we evaluate the impact of our interventions when only supervised fine-tuning only on the expanded language subset is feasible.

**UNIVERSAL tokenizer outperforms CLUSTER tokenizers by high margins in targeted adaptation for expanded language set.** Table 2b shows average win rates of UNIVERSAL and CLUSTER tokenizers for each geo-cluster. UNIVERSAL tokenizer achieves 10.2%, 15.7%, and 17.8% relative win rate gains over the CLUSTER-specific tokenizers for Euro, Asian, and ME-Indic clusters, respectively. In Figure 3a, we also plot individual language gains for the Euro cluster. UNIVERSAL consistently enables higher plasticity than CLUSTER tokenizer where the relative gains go up to 22.0% and 20.0% in Hindi and Farsi, respectively.

**UNIVERSAL tokenizer also provides large gains in targeted adaptation for fully unseen language set.** In the most extreme setting, we evaluate the benefits of our tokenizer intervention for adaptation to languages that are **fully unseen** in both tokenizer and pretraining. Figure 3b shows results on supervised fine-tuning experiments on 7 unseen languages. It is critical that all these languages are extremely under-resourced, and this adaptation is performed in a low-data environment, as this is representative of the constraints faced by developers in these languages.

**In the most extreme setting, UNIVERSAL tokenizer enables unseen language gains.** We find that UNIVERSAL tokenizer enables improvements over the CLUSTER tokenizer on the unseen languages with an average gain of 2.0% in win rates, where it goes up to 5.0% in Nepali. Given that the downstream performance is generally lower for these languages due to their absence in the tokenizer

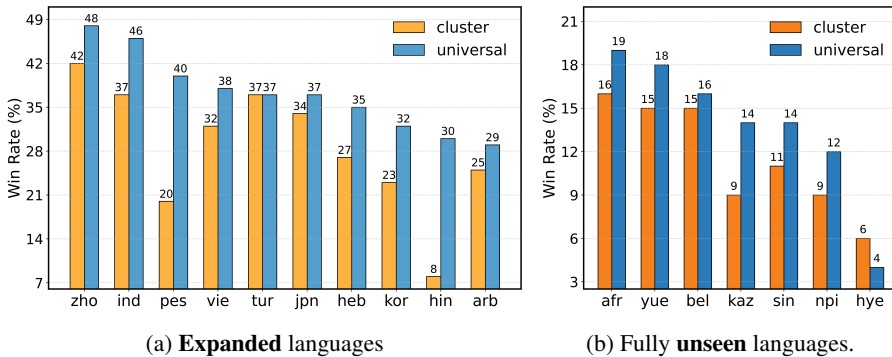

(a) **Expanded** languages

(b) Fully **unseen** languages.

Figure 3: Language-specific results after **targeted adaptation** through SFT for the Euro cluster model to (a) **expanded** languages and (b) fully **unseen** languages.

and pretraining, and also constraints on available data (only 15k per language), we find this to be a promising direction of future research and another reason to invest in more flexible tokenizer design.

In order to see whether the benefits of a universal tokenizer hold in the case of unseen adaptation languages that are higher-resourced, we augment the Nepali language with 21,000 samples from Sangraha, an instruction-style dataset for Indic languages (Khan et al., 2024). We replicate the unseen language SFT experiment, post-training both the European cluster models trained with the UNIVERSAL tokenizer and the CLUSTER tokenizer. We find that in the case of greater data availability, the advantage of using a universal tokenizer increases. On the smaller dataset, the UNIVERSAL tokenizer scores 3 points higher than the CLUSTER tokenizer. On the expanded data, the difference is 7%, with the UNIVERSAL showing win rate of 15.5%.

## 5 KEY DISCUSSIONS

### 5.1 COMPARISON WITH CROSS-LINGUAL VOCABULARY ADAPTATION

Cross-lingual vocabulary adaptation (CVA) (Yamaguchi et al., 2024b) aims to adapt the existing tokenizer and hence the token embeddings to the new languages through expansion or replacement after pretraining, and is a common approach for language adaptation. A more detailed overview of CVA can be found in Section 6. In this ablation, we ask: *how does the* UNIVERSAL *tokenizer compare with CVA for adapting new languages?*

To have a fully comparable setup with our body of experiments, we took the pretrained Euro cluster model that is trained with the CLUSTER tokenizer and replaced the tokenizer with the UNIVERSAL tokenizer. Token embeddings that are shared between CLUSTER and UNIVERSAL tokenizers are preserved, and new tokens are either randomly initialized by sampling from a normal distribution (`random`), or with the average of the shared embeddings (`mean`). After vocabulary replacement and token initialization, we follow the same continued pretraining described in Section 2.1.

The results for this ablation are given in Figure 4. We find that when randomly initializing the new tokens, CVA (tokenizer replacement) fails to achieve comparable performance even against the CLUSTER tokenizer, and significantly lags behind the UNIVERSAL tokenizer by 15.4% and 35.2% win rates for primary and expanded languages respectively. Notably, initializing the new tokens with the mean of the shared vocabulary (`mean` in Figure 4), outperforms random initialization. While tokenizer replacement (`mean`) is an improvement over the unadapted CLUSTER tokenizer by 12.8% relative increase in win rates for expanded languages, our UNIVERSAL tokenizer leads to better adaptation performance by 7% difference in average win rate (37.4% vs 30.4%). Interestingly, CVA (`mean`) achieves slightly higher performance in the primary languages by 2.1% average win rate. Overall, these results show that it is more effective to use a UNIVERSAL tokenizer from the start, rather than substituting it in after pretraining.

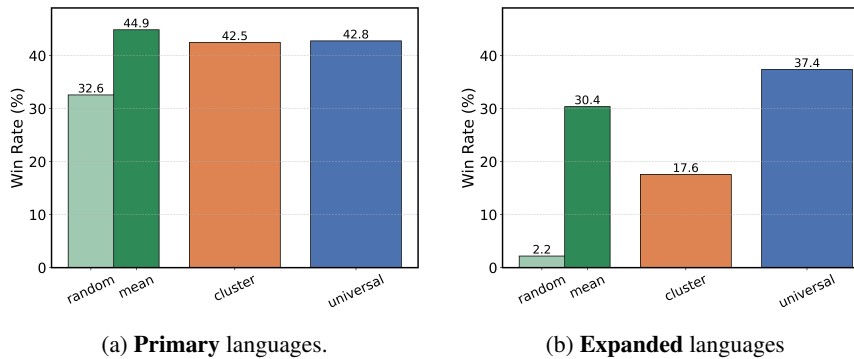

(a) **Primary** languages.     (b) **Expanded** languages

Figure 4: Win rates after continued pretraining, comparing Cross-lingual Vocabulary Adaptation (CVA) through tokenizer replacement.

## 5.2 ADAPTATION EFFICIENCY WITH THE UNIVERSAL TOKENIZER

In this ablation, we evaluate how much faster adaptation takes place with the UNIVERSAL tokenizer. Faster adaptation means much fewer resources necessary, namely cost, which is of great interest to practitioners who want to adapt an LLM to expanded language coverage.

To evaluate the adaptation speed, in comparison to CLUSTER tokenizer, we evaluated the intermediate checkpoints on the expanded languages during continued pretraining for Euro cluster models. Figure 1 shows average win rates on 10 expanded languages. As seen in the plot, in only 300 steps UNIVERSAL tokenizer reaches the level of performance that CLUSTER achieves at 2500 steps, showing +8x faster adaptation. Given that 300 steps correspond to nearly 150K samples (compared to 1.3M at 2500 steps), UNIVERSAL tokenizer requires also much less data to achieve the same performance with the end performance of the baseline, confirming the effectiveness of our proposal.

## 5.3 NECESSITY OF LARGE VOCABULARY SIZE

In the previous sections, we establish greater performance in multilingual plasticity with the UNIVERSAL compared to CLUSTER tokenizer. In this ablation, we ask: *What is the required vocabulary size for the* UNIVERSAL *tokenizer to avoid performance degradation on primary pretraining languages?*

To determine the optimal vocabulary size, we run additional pretraining experiments where we vary the vocabulary size from 100,000 tokens to 250,000 tokens while adjusting the model parameters so that the total number of trainable parameters remains constant. We evaluate the performance for the primary pretraining languages on Belebele. Results are shown in Figure 5a. The models trained with CLUSTER tokenizers don't vary much in performance, and surpass the UNIVERSAL tokenizer at small vocabulary sizes (100k and 175k). However, the UNIVERSAL tokenizer scales performance with the vocabulary size, and overtakes the CLUSTER tokenizer at 250k vocabulary size. Our findings are consistent with previous work that shows the benefits of large vocabularies (Tao et al., 2024; Huang et al., 2025), and suggests investment in universal tokenizers require a reallocation of weights to ensure a proper vocabulary budget. Based on this ablation, we use the vocabulary size of 250k in our main pretraining runs.

## 5.4 PRESENCE OF EXPANDED LANGUAGE SUBSET IN PRETRAINING

In the large and often noisy datasets used to train LLMs, there is often language contamination (Blevins & Zettlemoyer, 2022). Therefore, it can be difficult to claim a language is truly "new". In our final ablation, to test the robustness of our claims of plasticity under different assumptions of multilingual data presence, we evaluate 0%, 1%, and 5% proportion of expanded languages in pretraining for the European cluster.

Figure 5b shows that even in the most conservative case of 0% multilingual percentage for the new languages (the expanded subset), the UNIVERSAL tokenizer exhibits 12.8% gain in win rate

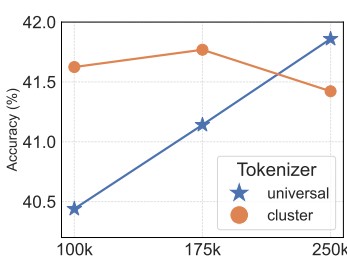 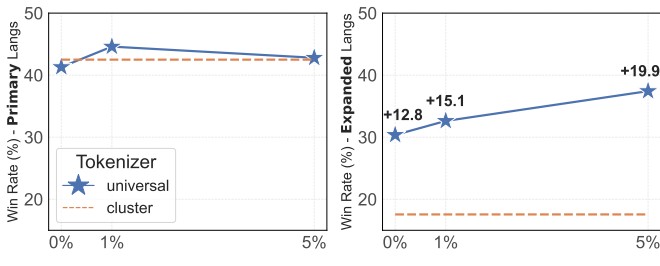

(a) Effect of vocabulary size.  (b) Effect of the expanded language data in pretraining

as compared to the CLUSTER tokenizer. Notably, increasing this percentage up to 5% does not hurt performance in primary pretraining languages, but increases adaptation performance on the expanded languages from 12.8% to 19.8% win rates.

## 6 RELATED WORK

There are a number of ways that language adaption of pretrained language models is commonly approached. In terms of additional training, continued pretraining (CPT) and supervised fine-tuning (SFT) are the most common methods. Continued pretraining involves extended training on the adaptation language corpora (Han & Eisenstein, 2019; Muller et al., 2021), but requires a significant amount of data, which may not be accessible for lower-resourced languages. Supervised fine-tuning is also a standard approach (Kumar et al., 2022; Adelani et al., 2021; Cahyawijaya et al., 2021), requiring less data than CPT, but possibly leading to catastrophic forgetting of capabilities from pretraining (Rolnick et al., 2019; Chaudhry et al., 2019). In particular, instruction fine-tuning is popular in order to impart instruction-following capabilities as well (Gala et al., 2024). Claims as to the advantages of one over the other are mixed- Ebrahimi & Kann (2021) find that in their setup, CPT is more effective than SFT, and Yong et al. (2023) find the opposite.

A primary challenge is unsupported scripts and languages in the tokenizer. Cross-lingual vocabulary adaptation (CVA) modifies the existing tokenizer to accommodate additional languages (Yamaguchi et al., 2024a), and requires continued pretraining in the target language to sufficiently adapt (Fujii et al., 2024). There are two common approaches to CVA – vocabulary expansion, where new tokens are added from the target and shared tokens are reused (Wang et al., 2020; Pfeiffer et al., 2021), or vocabulary replacement, where the vocabulary is entirely replaced. Embeddings corresponding to new tokens may be initialized randomly, using heuristics such as the average of some corresponding tokens in the original vocabulary (Minixhofer et al., 2022; Dobler & de Melo, 2023; Downey et al., 2023), or based on auxiliary models (Ostendorff & Rehm, 2023). Switching out the tokenizer is cumbersome; one possible method is by training a hypernetwork that maps vocab of the new tokenizer to existing embeddings (Minixhofer et al., 2025). This method requires continued training to close the performance gap, but even then doesn't surpass it. It has also been proposed to transliterate languages to Latin script to circumvent unsupported scripts (Muller et al., 2021), but this approach is limited by transliteration performance.

## CONCLUSION

In this work, we explore what cheap interventions in pretraining can increase plasticity in downstream optimization stages. We conduct an extensive study involving different tokenization strategies, three language adaptation strategies involving different assumptions about data access across 70 different languages. We find that in all cases, a model trained using a UNIVERSAL tokenizer with broad language coverage is able to adapt to languages outside of the primary pretraining set far better, with average win rate improvements up to 20.2% in continued pretraining and 17.8% in targeted adaptation to expanded languages. Even in the challenging low-data setting of completely unseen languages, the UNIVERSAL tokenizer shows gains up to 5%. At the same time, there is negligible performance impact to the primary pretraining languages. Investing in a massively multilingual tokenizer up-front pays off in language adaptation down the line.

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

## A  ADDITIONAL TOKENIZER DETAILS

For both cluster and universal tokenizer training data weighting, we use a fixed proportion of 30% for English due to much larger data volume and much higher diversity in available data. This also ensures a fair comparison between tokenizer weighting strategies.

We use the GPT-4o (`gpt4-o200k`) regex for pretokenization in all the tokenizers we trained.[1] Tokenizer training data is sampled from the pretraining data mixture based on the weights described in Section 2.3. We use the tokenizers [2] library to train all the BPE models. We set the `min_frequency` argument to 5 on the BPE trainer to control the minimum frequency to merge pairs and do not use normalizers. Finally, we sample 50GB of data to train all the tokenizers.

## B  ADDITIONAL TRAINING DETAILS

Our ablations are extensive and require a large amount of pretraining runs. Given the huge amount of compute required for pretraining, where training a 3.3B parameter model on 128 Nvidia H100 GPUs takes 11 hours, we focus on only 3.3 billion parameter language models for ablations. We train each base variant for 100 billion tokens using a total of 25,000 steps. Given the number of experiments we run, and the variety of factors we evaluate, this model size and amount of training steps is at the edge of what is computationally feasible at pretraining scale. Overall, the goal is not to emulate the settings of a full pretraining run, but to get sufficient signal about the relative merits of different approaches. In the continued pretraining strategy, we train for an additional 10.5B tokens and the targeted adaptation are done for 4 epochs over the respective datasets for each experiment.

**Hyperparameters** We performed a hyperparameter sweep on learning rate (LR), and used $2 \times 10^{-2}$ as the peak LR for all pretraining experiments. We use a batch size of 512, a sequence length of 8192, and a cosine learning rate scheduler with a warmup of 2500 steps. For language adaptation experiments after pretraining, we use a constant LR of $1 \times 10^{-4}$, corresponding to the end LR of the pretraining stage.

## C  ADDITIONAL EVALUATION DETAILS

**English-only evaluation.** Additionally, we also evaluate models on 11 English-only natural language inference and commonsense reasoning benchmarks: ARC-C and ARC-E (Chollet, 2019), BoolQ (Clark et al., 2019), CommonsenseQA (Talmor et al., 2019), Hellaswag (Zellers et al., 2019), MMLU (Hendrycks et al., 2021), OpenBookQA (Mihaylov et al., 2018), PIQA (Bisk et al., 2020), SIQA (Sap et al., 2019), TruthfulQA (Lin et al., 2022), and WinoGrande (Sakaguchi et al., 2019).

We include task-specific evaluations (both multilingual and English-only) to understand the relative merit of different design choices. Typically pretrained models do not perform well at downstream tasks at this point in training, as the models have not yet been optimized for instruction following (Wang et al., 2022; Üstün et al., 2024; Aakanksha et al., 2024), or aligned using reinforcement learning (Ahmadian et al., 2024; Dang et al., 2024). Hence, we do not expect state-of-the-art performance, but rather evaluate the relative signal of different variants.

## D  ADDITIONAL RESULTS

### D.1  ADDITIONAL PRETRAINING RESULTS

Figure 6 shows pretraining results on the primary language subset (Euro cluster), measured in Belebele, throughout the pretraining run.

---

[1] https://github.com/openai/tiktoken/blob/4560a8896f5fb1d35c6f8fd6eee0399f9a1a27ca/tiktoken_ext/openai_public.py#L95

[2] https://github.com/huggingface/tokenizers

| | bul | cat | ces | dan | deu | ell | est | eus | fin | fra | hrv | hun | ita | lit |
|---|---|---|---|---|---|---|---|---|---|---|---|---|---|---|
| UNIFORM | 41.0 | 43.7 | 42.7 | 41.5 | 43.6 | 41.2 | 35.7 | 38.4 | 34.6 | 45.0 | 42.1 | 36.6 | 40.6 | 41.0 |
| UNIVERSAL | 42.1 | 46.5 | 45.5 | 41.1 | 44.3 | 42.7 | 37.9 | 40.6 | 32.4 | 45.3 | 42.6 | 37.9 | 40.1 | 43.1 |
| | (+1.1) | (+2.8) | (+2.8) | (−0.4) | (+0.7) | (+1.5) | (+2.2) | (+2.2) | (−2.2) | (+0.3) | (+0.5) | (+1.3) | (−0.5) | (+2.1) |

| | lvs | nld | nob | pol | por | ron | rus | slk | slv | spa | srp | swe | ukr | Average |
|---|---|---|---|---|---|---|---|---|---|---|---|---|---|---|
| UNIFORM | 42.3 | 42.7 | 42.3 | 38.4 | 41.0 | 40.8 | 41.0 | 42.4 | 39.5 | 42.5 | 42.4 | 42.9 | 40.9 | 41.0 |
| UNIVERSAL | 40.5 | 42.7 | 42.8 | 39.8 | 43.8 | 41.2 | 41.4 | 43.5 | 41.8 | 43.8 | 43.4 | 43.4 | 40.0 | 41.9 |
| | (−1.8) | (+0.0) | (+0.5) | (+1.4) | (+2.8) | (+0.4) | (+0.4) | (+1.1) | (+2.3) | (+1.3) | (+1.0) | (+0.5) | (−0.9) | (+0.9) |

Table 3: Comparison of UNIVERSAL vs. UNIFORM tokenizer performance on Belebele, when used for pretraining of the Euro cluster model.

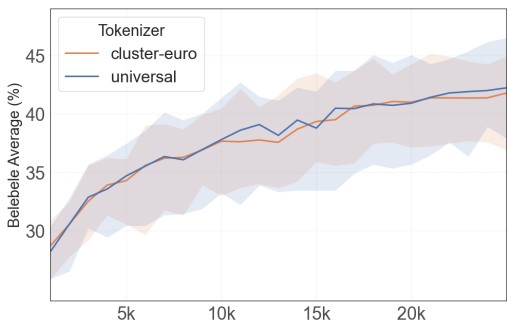

Figure 6: Average performance on primary languages during pretraining (Euro), measured in Belebele. Shaded areas indicate results across languages for each tokenizer. UNIVERSAL tokenizer shows nearly the same performance with CLUSTER tokenizer throughout the training, suggesting a similar performance also in much longer pretraining runs.

## D.2 EXPANDED LANGUAGE ADAPTATION RESULTS

### D.2.1 CONTINUED PRETRAINING

Table 4 presents the win rates by language after continued pretraining, divided into PRIMARY and EXPANDED languages

### D.2.2 TARGETED ADAPTATION

Table 4 presents the win rates by language after continued pretraining, divided into PRIMARY and EXPANDED languages

## D.3 JUDGE PROMPT FOR WIN RATES

---

**¡system_prompt¿**
You are a helpful following assistant whose goal is to select the preferred (least wrong) output for a given instruction.

**¡user_prompt¿**
Which of the following answers is the best one for the given instruction? A good answer should follow these rules:
1) It should have correct reasoning,"
2) It should answer the request in the instruction,
3) It should be factually correct and semantically comprehensible,
4) It should be grammatically correct and fluent.

Instruction: instruction
Answer (A): completion_a
Answer (B): completion_b

FIRST provide a concise comparison of the two answers. If one answer is better, explain which you prefer and why. If both answers are identical or equally good or bad, explain why. SECOND, on a new line, state exactly one of 'Answer (A)' or 'Answer (B)' or 'TIE' to indicate your choice of preferred response.
Your response should use the format: Comparison: ¡concise comparison and explanation¿ Preferred: ¡'Answer (A)' or 'Answer (B)' or 'TIE'¿.

---

| | | Asian | | | ME-Indic | |
|---|---|---|---|---|---|---|
| | | CLUSTER | UNIVERSAL | | CLUSTER | UNIVERSAL |
| PRIMARY | ind | 56.5 | 46.5 | arb | 37.5 | 43.0 |
| | jpn | 52.0 | 40.5 | heb | 38.0 | 39.0 |
| | kor | 46.0 | 40.0 | hin | 41.5 | 38.0 |
| | vie | 49.5 | 39.5 | pes | 46.0 | 42.5 |
| | zho | 52.5 | 45.0 | tur | 35.5 | 38.5 |
| EXPANDED | arb | 16.0 | 27.0 | deu | 18.0 | 35.5 |
| | deu | 25.5 | 27.5 | ell | 10.6 | 29.5 |
| | ell | 9.0 | 29.0 | fra | 28.0 | 43.0 |
| | fra | 31.5 | 35.5 | ind | 24.5 | 46.5 |
| | heb | 6.1 | 24.5 | jpn | 20.5 | 46.0 |
| | hin | 5.1 | 27.5 | kor | 19.0 | 37.5 |
| | pes | 13.5 | 33.5 | rus | 20.5 | 41.0 |
| | rus | 17.5 | 38.0 | spa | 35.5 | 46.5 |
| | spa | 38.0 | 43.0 | vie | 16.0 | 38.0 |
| | tur | 13.0 | 27.5 | zho | 35.5 | 54.0 |

Table 4: Full win rate results by language for language adaptation through continued pretraining.

| | Asian | | | ME-Indic | |
|---|---|---|---|---|---|
| | CLUSTER | UNIVERSAL | | CLUSTER | UNIVERSAL |
| arb | 16.5 | 30.5 | deu | 20.5 | 35.5 |
| deu | 20.0 | 30.5 | ell | 19.1 | 32.5 |
| ell | 13.1 | 32.0 | fra | 27.5 | 39.0 |
| fra | 27.5 | 37.0 | ind | 20.0 | 46.5 |
| heb | 12.1 | 32.0 | jpn | 16.5 | 41.5 |
| hin | 9.1 | 25.8 | kor | 16.5 | 37.0 |
| pes | 17.0 | 41.5 | rus | 22.5 | 39.0 |
| rus | 20.0 | 37.5 | spa | 39.5 | 46.5 |
| spa | 33.0 | 43.5 | vie | 16.5 | 43.0 |
| tur | 15.5 | 30.0 | zho | 34.5 | 50.5 |

Table 5: Targeted language adaptation win rates

# E    COMPRESSION RATIO

In order to intrinsically evaluate tokenizer quality, we measure compression ratio as compared to the publicly-available multilingual tokenizer used in Command-A (Cohere et al., 2025). Compression measures how efficiently data is represented in terms of size (in bytes), and BPE optimizes for this condition (Gage, 1994). Compression ratio compares compression values between tokenizers, and since lower compression is desirable, a compression ratio below 1 indicates that a tokenizer has more favorable compression. Previous work shows that compression correlates well with model performance, especially for generative tasks (Goldman et al., 2024; Gallé, 2019), although lower compression is not a sufficient condition for a better tokenizer (Schmidt et al., 2024). However, long sequence lengths are one of the ways in which inequitable treatment of languages begins at the tokenizer (Velayuthan & Sarveswaran, 2025; Ahia et al., 2023), and is therefore an important measure to consider along with downstream evaluations.

## E.1    IMPACT OF TOKENIZER LANGUAGE WEIGHTING ON COMPRESSION RATIO

As a baseline, we evaluate uniform language weighting and compare it with our tokenizer where we use both data distribution and language bucketing strategies in conjunction. Compression ratios are computed against the multilingual tokenizer of open weight Command-A (Cohere et al., 2025) model on the test split of FineWeb-2 (Penedo et al., 2024).

Figure 7 shows the comparison. As seen, our tokenizer, which uses a special weighting, leads to better compression than the uniform baseline. Note that both of these tokenizers achieve overall higher compression performance than Command-A since they are trained with larger language coverage.

## E.2    IMPACT OF COMPRESSION RATIO ON DOWNSTREAM PERFORMANCE

Figure 8 explores the relationship between compression ratio and win rates for European cluster models trained with the UNIVERSAL and CLUSTER tokenizer on primary and expanded languages. The expansion languages exhibit large compression ratios with the CLUSTER, all over 1, which indicates that compression in that language is worse than the comparison tokenizer. At the same

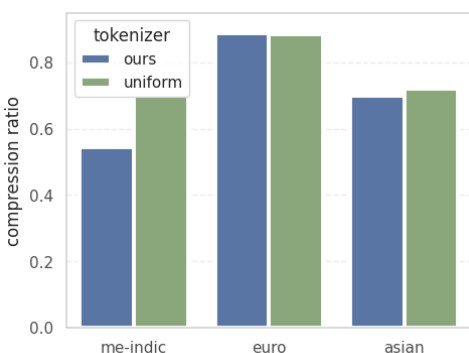

Figure 7: Compression ratios for our tokenizer and the baseline uniform tokenizer. Our tokenizer uses a special weighting that leverages training data distribution and the language grouping (§ 2.3), leading to a better compression (lower is better).

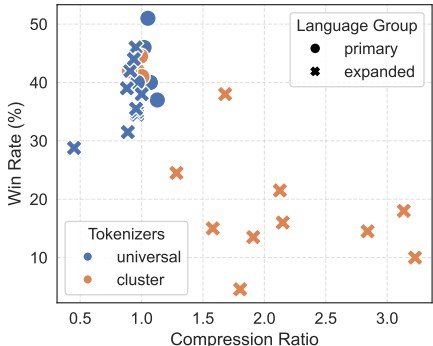

Figure 8: Adaptation results per language in Euro cluster together with tokenizers' compression ratio. While UNIVERSAL tokenizer enables better compression, especially for the expanded language subset, hence, better downstream performance, CLUSTER tokenizer fails to represent these languages, leading to lower adaptation results.

time, the win rates for those languages are lower than the primary languages, which also have a lower compression ratio. The UNIVERSAL tokenizer, however, shows relatively high win rates and low compression ratios for both primary and expansion languages. This result corroborates the relationship between compression ratio and downstream performance, and provides an additional dimension for language plasticity of the UNIVERSAL tokenizer.

## F  ADDITIONAL RELATED WORK

Tokenization remains a key challenge for multilingual models, particularly due to inefficiencies and disparities across scripts. Petrov et al. (2023) show that many standard or English-centric tokenizers disproportionately fragment non-Latin scripts, sometimes producing up to 15 times more tokens than for equivalent English content. These imbalances can lead to practical consequences such as increased API usage costs (Ahia et al., 2023; Petrov et al., 2023), longer inference times (Hofmann et al., 2022; Sun et al., 2023), and reduced usable context window for non-English languages (Velayuthan & Sarveswaran, 2025; Ahia et al., 2023), and degraded downstream task performance (Goldman et al., 2024; Gow-Smith et al., 2022; Fujii et al., 2023). Moreover, inefficient tokenization for non-English languages can inflate training costs by up to 68% (Ali et al., 2024).

To support multilinguality, the mT5 model introduced a 250k SentencePiece vocabulary with byte-fallback, pretrained on 101 languages using temperature-based sampling to balance high- and low-resource languages (Xue et al., 2021). Additionally, recent work has explored tokenization approaches tailored to better reflect the structure of diverse writing systems. For example, Grapheme-

aware tokenization, using Unicode grapheme clusters as atomic units via methods like Grapheme Pair Encoding (GPE) (Velayuthan & Sarveswaran, 2025) or MYTE, a segmentation strategy motivated by morphemes (Limisiewicz et al., 2024), offers improved representation for complex scripts.

## G Limitations

**Language coverage.** Our experiments involve 69 languages covering a diverse range of languages and scripts, where we systematically investigate the impact of multilingual data on tokenizer training. Although we use a comprehensive list of languages, there are many more languages in the world, which requires attention of the research community. We hope our work encourages even broader language coverage in state-of-the-art language models.

**Tokenization algorithm.** In this work, we focused only on the BPE algorithm for tokenizer training, which is the most widely used method for language models. This choice was dictated by the high computational cost of each ablation, which required significant compute resources. However, we believe our findings on multilingual coverage would apply to the other tokenizers, such as Unigram tokenizer (Kudo, 2018) or byte or character-level tokenization (Xue et al., 2022; Clark et al., 2022). We leave this exploration to future research.

**Model size.** All of our pretraining experiments were conducted on 3.3B parameter models with a 100B token budget, which is already an enormous undertaking for resources and compute costs. Given that our results hold for this scale, we anticipate they would also apply to larger models and token budgets, as supported by previous research (Biderman et al., 2023; Longpre et al., 2024; Aryabumi et al., 2024).

## H Languages

Table 6: Pretraining languages, including pretraining cluster assignment. Languages with a checkmark in the post-training column but without cluster assignment are used as unseen adaptation languages.

| ISO Code | Language | Script | Family | Subgrouping | Resources | Cluster | In Post-Training |
|---|---|---|---|---|---|---|---|
| afr | Afrikaans | Latin | Indo-European | Germanic | Mid | - | ✓ |
| ara | Arabic | Arabic | Afro-Asiatic | Semitic | High | Me-Indic | ✓ |
| amh | Amharic | Ge'ez | Afro-Asiatic | Semitic | Low | - | ✗ |
| bel | Belarusian | Cyrillic | Indo-European | Balto-Slavic | Mid | - | ✓ |
| ben | Bengali | Bengali | Indo-European | Indo-Aryan | Mid | Me-Indic | ✗ |
| bul | Bulgarian | Cyrillic | Indo-European | Balto-Slavic | Mid | Euro | ✗ |
| cat | Catalan | Latin | Indo-European | Italic | High | Euro | ✗ |
| ces | Czech | Latin | Indo-European | Balto-Slavic | High | Euro | ✓ |
| cym | Welsh | Latin | Indo-European | Celtic | Low | Euro | ✗ |
| dan | Danish | Latin | Indo-European | Germanic | Mid | Euro | ✗ |
| deu | German | Latin | Indo-European | Germanic | High | Euro | ✓ |
| ell | Greek | Greek | Indo-European | Graeco-Phrygian | Mid | Euro | ✓ |
| eng | English | Latin | Indo-European | Germanic | High | Euro | ✓ |
| est | Estonian | Latin | Uralic | Finnic | Mid | Euro | ✗ |
| eus | Basque | Latin | Basque | - | High | Euro | ✗ |
| fil | Filipino | Latin | Austronesian | Malayo-Polynesian | Mid | Asian | ✗ |
| fin | Finnish | Latin | Uralic | Finnic | Mid | Euro | ✗ |
| fra | French | Latin | Indo-European | Italic | High | Euro | ✓ |
| gla | Scottish Gaelic | Latin | Indo-European | Celtic | Low | Euro | ✗ |
| gle | Irish | Latin | Indo-European | Celtic | Low | Euro | ✗ |
| glg | Galician | Latin | Indo-European | Italic | Mid | Euro | ✗ |
| guj | Gujarati | Gujarati | Indo-European | Indo-Aryan | Low | Me-Indic | ✗ |
| heb | Hebrew | Hebrew | Afro-Asiatic | Semitic | Mid | Me-Indic | ✓ |
| hin | Hindi | Devanagari | Indo-European | Indo-Aryan | High | Me-Indic | ✓ |
| hrv | Croatian | Latin | Indo-European | Balto-Slavic | High | Euro | ✗ |
| hun | Hungarian | Latin | Uralic | - | High | Euro | ✗ |
| hye | Armenian | Armenian | Indo-European | Armenic | Low | - | ✓ |
| ibo | Igbo | Latin | Atlantic-Congo | Benue-Congo | Low | - | ✗ |
| ind | Indonesian | Latin | Austronesian | Malayo-Polynesian | Mid | Asian | ✓ |
| ita | Italian | Latin | Indo-European | Italic | High | Euro | ✓ |
| jav | Javanese | Latin | Austronesian | Malayo-Polynesian | Low | Asian | ✗ |
| jpn | Japanese | Japanese | Japonic | Japanesic | High | Asian | ✓ |
| kaz | Kazakh | Cyrillic | Turkic | Common Turkic | Mid | - | ✓ |
| khm | Khmer | Khmer | Austroasiatic | Khmeric | Low | Asian | ✗ |
| kor | Korean | Hangul | Koreanic | Korean | Mid | Asian | ✓ |

| ISO Code | Language | Script | Family | Subgrouping | Resources | Cluster | In Post-Training |
|---|---|---|---|---|---|---|---|
| lao | Lao | Lao | Tai-Kadai | Kam-Tai | Low | Asian | ✗ |
| lav | Latvian | Latin | Indo-European | Balto-Slavic | Mid | Euro | ✗ |
| lit | Lithuanian | Latin | Indo-European | Balto-Slavic | Mid | Euro | ✗ |
| mlt | Maltese | Latin | Afro-Asiatic | Semitic | Low | Me-Indic | ✗ |
| msa | Malay | Latin | Austronesian | Malayo-Polynesian | Mid | Asian | ✗ |
| mya | Burmese | Myanmar | Sino-Tibetan | Burmo-Qiangic | Low | Asian | ✗ |
| nep | Nepali | Devanagari | Indo-European | Indo-Aryan | Low | - | ✓ |
| nld | Dutch | Latin | Indo-European | Germanic | High | Euro | ✓ |
| nor | Norwegian | Latin | Indo-European | Germanic | Low | Euro | ✗ |
| pan | Punjabi | Gurmukhi | Indo-European | Indo-Aryan | Low | Me-Indic | ✗ |
| pes | Persian | Arabic | Indo-European | Iranian | High | Me-Indic | ✓ |
| pol | Polish | Latin | Indo-European | Balto-Slavic | High | Euro | ✓ |
| por | Portuguese | Latin | Indo-European | Italic | High | Euro | ✓ |
| ron | Romanian | Latin | Indo-European | Italic | Mid | Euro | ✓ |
| rus | Russian | Cyrillic | Indo-European | Balto-Slavic | High | Euro | ✓ |
| sin | Sinhala | Sinhala | Indo-European | Indo-Aryan | Low | - | ✓ |
| slk | Slovak | Latin | Indo-European | Balto-Slavic | Mid | Euro | ✗ |
| slv | Slovenian | Latin | Indo-European | Balto-Slavic | Mid | Euro | ✗ |
| spa | Spanish | Latin | Indo-European | Italic | High | Euro | ✓ |
| srp | Serbian | Cyrillic | Indo-European | Balto-Slavic | High | Euro | ✗ |
| swa | Swahili | Latin | Atlantic-Congo | Benue-Congo | Low | - | ✗ |
| swe | Swedish | Latin | Indo-European | Germanic | High | Euro | ✗ |
| tam | Tamil | Tamil | Dravidian | South Dravidian | Mid | Me-Indic | ✗ |
| tel | Telugu | Telugu | Dravidian | South Dravidian | Low | Me-Indic | ✗ |
| tha | Thai | Thai | Tai-Kadai | Kam-Tai | Mid | Asian | ✗ |
| tur | Turkish | Latin | Turkic | Common Turkic | High | Me-Indic | ✓ |
| ukr | Ukrainian | Cyrillic | Indo-European | Balto-Slavic | Mid | Euro | ✓ |
| urd | Urdu | Arabic | Indo-European | Indo-Aryan | Mid | Me-Indic | ✗ |
| vie | Vietnamese | Latin | Austroasiatic | Vietic | High | Asian | ✓ |
| xho | Xhosa | Latin | Atlantic-Congo | Benue-Congo | Low | - | ✗ |
| yor | Yorùbá | Latin | Atlantic-Congo | Benue-Congo | Low | - | ✗ |
| yue | Cantonese | Han | Sino-Tibetan | Sinitic | Low | - | ✓ |
| zho | Mandarin Chinese | Han | Sino-Tibetan | Sinitic | High | Asian | ✓ |
| zul | Zulu | Latin | Atlantic-Congo | Benue-Congo | Low | - | ✗ |

