# OpenReview forum: "One Tokenizer To Rule Them All: Emergent Language Plasticity via Multilingual Tokenizers"
_ICLR.cc/2026/Conference — Submitted to ICLR 2026_

### Official Review · Reviewer_JwKn · 2025-10-25

**Soundness:** 2
**Presentation:** 2
**Contribution:** 1
**Rating:** 0
**Confidence:** 5

**Summary:**

In this work, the authors train LLMs using larger-than-usual tokenizers, and show that this added capacity does not benefit much popular languages, but can benefit greatly less represented languages if they are upsampled properly. They also show that adding tokens post-training is not fully as effective as training with them from the start, if done with baseline methods (random init, mean embedding init). These initializations are however far from the existing SOTA, and it's not clear that this conclusion would still hold with more recent initialization strategies such as (FOCUS, WECHSEL, or Trans-Tokenization).

**Strengths:**

* The authors aim to solve a very important problem, the poor performance of language models in less-represented languages.
* The proposed solution is reasonable, although this type of language reweighting is fairly standard for all LLM tokenizers these days.
* The article is globally well-written and accessible.

**Weaknesses:**

* The paper only shows that increasing the vocab size to 250k yields little to no benefit once the target language is properly modeled (e.g. about 100k tokens), while using that added capacity for other languages is helpful for later finetuning the model on those languages. This is pretty obvious, and well-documented. Even the first multilingual llama paper discusses this, to some extent. The reason why tokenizer size is usually kept smaller than 250k is the cost involved in a larger modeling head + softmax, which is not discussed at all in the paper. It is somehow assumed that chaging the vocab size is a free operation, which isn't true at all in practice. I would refer to Apple's paper on Hierachical softmax for possible ways to alleviate this issue.
* Some of the languages chosen as "completely unseen" are not reasonable choices for this category. For example, the first one (written Afrikaans) is extremely similar to written Dutch, which is a seen language. The second one (Belarussian) is very close from Russian and Ukrainian, especially in writing. Those languages will naturally vastly benefit from an appropriate tokenizer with their tokens trained on similar languages. This isn't a surprising or novel finding.
* More globally, no tokenizer is universal. The claim in this paper to have invented a universal tokenizer is baseless, as for instance the "universal" tokenizer will not be able to perform well on languages like Armenian if they weren't seen during the tokenization training, since the language uses unique code points not shared with other languages, which would then be decomposed in byte sequences by BPE. Any language not seen during the tokenizer training would suffer the same issue, and not all languages are in the training set.

**Questions:**

* Can you justify your statement that `Overall, these results show that it is more effective to use a UNIVERSAL tokenizer from the start,
 rather than substituting it in after pretraining` in the light of the significant cost caused by a large tokenizer during training of the model?
* Why did you only compare your method with pretty basic baseline CVA techniques, rather than more recent ones which have been shown to provide huge benefits over these baselines at none to very low training costs?

---

> ### Author Response · Authors · 2025-11-21
> **Official Comment by Authors**
>
> We thank **Reviewer JwKn** for recognizing the importance of the problem we have chosen to tackle as well as the clarity of our writing, and are grateful for the opportunity to respond to the noted concerns.
>
> > Can you justify your statement that “Overall, these results show that it is more effective to use a UNIVERSAL tokenizer from the start, rather than substituting it in after pretraining” in the light of the significant cost caused by a large tokenizer during training of the model?
>
> ~250k has been widely used as a vocabulary size in various leading models at the same scale we use, notably the Gemma model series [1,4,5] on the 2b and 4b scale. There have been older models such as PaLM-8B [2] as well. Modern LLMs with a vanilla softmax LM head, especially with long context windows, use techniques such as tied embeddings to reduce memory pressure both in training and inference [4,1], and specialized kernels [6,7] to fuse the LM projection and cross-entropy which avoids materializing the full logits during training. On the inference side, sorting large vocabs becomes a bottleneck, but there are recent works performing sorting-free sampling for large vocabularies [8]. In general by construction of BPE, bigger vocabularies will have better compression. [9] argues that compression is intimately related to the very objective LMs optimize, and shows empirically correlation to model performance.
>
> [1] “Gemma3 Technical Report.” arXiv:2503.19786
> [2] “PaLM: Scaling Language Modeling with Pathways.” (Journal of Machine Learning Research 2023)
> [3]  Unpacking Tokenization: Evaluating Text Compression and its Correlation with Model Performance (ACL 2024)
> [4] Gemma 2: Improving Open Language Models at a Practical Size arXiv:2408.00118
> [5] Gemma: Open Models Based on Gemini Research and Technology  arXiv:2403.08295
> [6] Liger-Kernel: Efficient Triton Kernels for LLM Training (ICML - CODEML workshop 2025)
> [7] Cut Your Losses in Large-Vocabulary Language Models (ICLR 2025)
> [8] FlashInfer sampling: https://flashinfer.ai/2025/03/10/sampling.html
>
> > Why did you only compare your method with pretty basic baseline CVA techniques, rather than more recent ones which have been shown to provide huge benefits over these baselines at none to very low training costs?
>
> Random and mean initialization are fairly standard CVA techniques and serve as a reasonable baseline [1, 2, 3]. In fact, FOCUS initializes embeddings from the mean of shared tokens [4]. We recognize that there are other more advanced techniques than the baselines we tested, and we will expand the CVA section of the related work to outline these techniques.
> To clarify, we do not claim that the universal tokenizer is the most effective method to enable vocabulary adaptation, but it is a low-cost alternative in the pretraining design that does not require complex intervention in posttraining for language adaptation.
>
> [1] “Mini-Model Adaptation: Efficiently Extending Pretrained Models to New Languages via Aligned Shallow Training.” (ACL 2023)
> [2] “OFA” (NAACL 2024)
> [3] “As Good as New. How to Successfully Recycle English GPT-2 to Make Models for Other Languages.” (ACL 2021)
> [4] “FOCUS” (EMNLP 2023)
>
> > More globally, no tokenizer is universal. The claim in this paper to have invented a universal tokenizer is baseless
>
> At no point do we claim to have invented a truly universal tokenizer – we even test unseen languages which are not present in the tokenizer training. UNIVERSAL is used within our paper as a relative term indicating broader language coverage than the pretraining focus languages. We believe that this is a very clear point in the paper but we’re happy to make it more pronounced.

---

> > ### Comment · Reviewer_JwKn · 2025-11-25
> >
> > Thank you for your thoughtful reply.
> >
> > Unfortunately, I don't think that this response sufficiently addresses the concerns raised.
> >
> > 1. To support the claim that making the tokenizer larger than needed for the main languages is cheaper than vocabulary transfer, you have to take into account the additional cost of training with a larger tokenizer, and show that using this additional cost later in the post-training does not yield better results. At the moment, the paper neither quantifies the effect of a larger tokenizer, nor attempts to show that vocabulary adaptation is not sufficient to address the stated problem without or with minimal compute cost.
> >
> > 2. While I am sympathetic to the idea that `At no point do we claim to have invented a truly universal tokenizer`, the article says multiple times `the UNIVERSAL tokenizer` which leads to confusion. An attempt to address this concern by renaming the tokenizer would have been considered a step in the right direction.
> >
> > 3. I will also note that the reply does not address the concerns related to the choice of "unseen" languages, which are extremely related and likley share many tokens with seen languages. In my opinion, the main reason benefits are visible is that many tokens are shared between languages in the seen and unseen categories with identifical or similar meanings, and this hypothesis has not been tested using a proper test.
> >
> > As a result of the above, I did not upgrade my rating for this paper.

---

> ### Author Response · Authors · 2025-12-03
> **Author Response**
>
> We are appreciative of Reviewer **JwKn**’s responsiveness and their openness to discuss further, but we regret that the rebuttal period has been frozen.
>
> 1. To address vocabulary size, models are parameter-matched across all vocabulary sizes in our experiments. The benefit of a larger vocabulary size is shown in section 5.3, where we evaluate the pretrained models on primary languages to ensure that the performance of primary languages doesn’t suffer as more languages are included in tokenizer training. We demonstrate the advantage of the universal tokenizer against vocabulary adaptation in section 5.1, where the universal tokenizer yields 7% higher winrate on expanded languages, against our CVA baseline. Readers are free to assess whether or not the significant advantage of a larger vocabulary size is worth negligible additional costs in training with a larger tokenizer, and we have outlined why these costs are negligible in our previous response. We hope our work provides evidence to help inform that decision.
>
> 2. As this is causing significant confusion, we will include a disclaimer clarifying that Universal is just a relative term. Changing the name would require substantive revisions to the paper. This hasn’t been a source of confusion otherwise, but we want to make our work as clear as possible.
>
> 3. Unseen languages were selected to optimize for diversity, such as Sinhala, which is written in an unseen script. We see a consistent advantage with the universal tokenizer, even in Sinhala. Of course, these choices are made within the bounds of data availability. We are happy to include compression ratios, to show that the compression of these tokenizers on unseen languages is not as effective as languages included in training. We agree that shared tokens are almost certainly one of the reasons that we observe the enhanced plasticity of models trained with the universal tokenizer. We show that having as many languages as possible in the tokenizer does not have a diminishing gain (even if they are extremely underrepresented during pretraining), and leads to much better adaptation. We apologize for the oversight of not addressing this point in our earlier response.

---

### Official Review · Reviewer_UXUT · 2025-10-27

**Soundness:** 3
**Presentation:** 4
**Contribution:** 3
**Rating:** 8
**Confidence:** 4

**Summary:**

This paper investigates how tokenizer design affects the language plasticity of multilingual LLMs -- i.e., their ability to adapt to new or unseen languages after pretraining.
The authors propose a "Universal" Tokenizer, trained on a much broader set of languages than those used during pretraining, as a low-cost intervention to improve post-training adaptability.
In their setup, the authors consider 62 languages, which are broken up into three clusters: (1) European languages, (2) Asian languages, and (3) Middle-Eastern and Indic languages.
The authors then conducted controlled experiments by pretraining models in the "primary" cluster, while using the other clusters as "expanded" subsets (there are also 7 unseen languages) for plasticity adaptation evaluation with multiple strategies (continued pretraining, targeted fine-tuning, and unseen-language adaptation).
The authors show that the universal tokenizer enables higher win rates in adaptation to expanded languages and unseen ones, while maintaining comparable performance on pretraining languages.
Additionally, the experiments indicate that the universal tokenizer yields faster adaptation and better compression efficiency.
The authors also show that the universal tokenizer outperforms cross-lingual vocabulary adaptation.

**Strengths:**

- The experiments are very extensive and well-controlled (69 languages, 3 language clusters + 7 unseen languages, multiple adaptation regimes).

- There is clear evidence of improvement in plasticity and adaptation efficiency without trade-offs on primary languages.

- The authors provide clear evidence that a universal tokenizer is a simple, yet scalable and low-cost method to enhance multilingual coverage.

**Weaknesses:**

- The use of LLM-as-a-Judge for open-ended generation is reasonable but subjective. Given that 15 adaptation languages are evaluated, including even a small-scale human verification subset would strengthen the credibility of the win-rate results.

- Experiments are conducted on a 3.3B model. Since multilingual capability often scales with model size and inference budget [1], it remains unclear whether the Universal Tokenizer's benefits persist, or diminish, at larger scales.

I understand the difficulty of evaluating open-ended generation. The authors use LLM-as-a-Judge and report the win rates. Since the authors use 15 adaptation languages in this test, I think it would be more convincing if a small-scale human evaluation is included.

- The model scale (3.3B) is not very large. Since the model capability (including multilinguality) has shown to be improved quite a bit with scaling up when giving enough inference budget [1], I am wondering if the benefit of using universal tokenizer decreases when the model size scales up.

- The CVA comparisons use relatively simple mean and random initialization methods. Stronger baselines (e.g., from [2, 3, 4]) could provide a fairer reference point, though this is not critical given the paper's focus on tokenizer design rather than adaptation algorithms.


[1] https://arxiv.org/abs/2505.05408
[2] https://arxiv.org/abs/2112.06598
[3] https://arxiv.org/abs/2305.14481
[4] https://arxiv.org/abs/2311.08849

**Questions:**

- Could you provide a quantitative breakdown of token overlap ratios between Universal and Cluster tokenizers across scripts?

- Have you considered multilingual reasoning tasks (e.g., MGSM) to test if improved plasticity translates to reasoning?

- Since the Universal Tokenizer covers languages unseen during pretraining, many subword embeddings might be undertrained or even untrained, especially for low-resource or distinct-script languages. Can the authors provide intuition for why these embeddings still facilitate adaptation in post-training stages?

- Relatedly, could you quantify the proportion of subwords that are (1) untrained and (2) undertrained after pretraining? This would help explain the mechanism behind the Universal Tokenizer's effectiveness.

- OFA [1] is another wise embedding initialization strategy for CVA; including this could further strengthen the completeness of the related work..

[1] https://arxiv.org/abs/2311.08849

---

> ### Author Response · Authors · 2025-11-21
> **Official Comment by Author**
>
> We warmly thank **Reviewer UXUT** for their positive feedback, emphasizing the extensiveness of our experiments, and strength and clarity of evidence.
>
> > Could you provide a quantitative breakdown of token overlap ratios between Universal and Cluster tokenizers across scripts?
>
> The following is the ratio of token overlap between the cluster and universal tokenizers in a given script to the total number of tokens fully in that script.
> |           | **European** |   |   | **Asian**  |   |   |   |   |   |   |   | **Middle Eastern & Indic**  |   |   |   |   |   |   |   |
> | :-------: | :---: | :------: | :------: | :------: | :------: | :------: | :------: | :------: | :------: | :------: | :------: | :------: | :------: | :------: | :------: | :------: | :------: | :------: | :------: |
> | Script    | Latin | Cyrillic | Greek | Hanzi | Hiragana | Khmer | Myanmar | Katakana | Thai | Hangul | Lao | Arabic | Telugu | Tamil | Hebrew | Devanagari | Gurmukhi | Bengali | Gujarati |
> | Universal | 0.78  | 0.51     |  0.89  | 0.98  | 0.95     | 0.92  | 0.75    | 0.87     | 0.76 | 0.96   | 0.86 | 0.91   | 0.88   | 0.93  | 0.14    | 0.88       | 0.92     | 0.88    | 0.87     |
> | Cluster   | 0.64  | 0.27     | 0.56  | 0.28  | 0.27     | 0.32  | 0.35    | 0.27     | 0.33 | 0.31   | 0.29 | 0.39   | 0.34   | 0.36  | 0.26    | 0.34       | 0.34     | 0.35    | 0.33     |
>
> We will include an analysis of token overlap ratios and shared tokens across languages and scripts between tokenizers in the appendix.
>
> > Have you considered multilingual reasoning tasks (e.g., MGSM) to test if improved plasticity translates to reasoning?
>
> A variety of evaluation tasks is certainly important – we’ve included multiple kinds of evaluation tasks (namely Belebele, machine translation, Global MMLU, and open-ended questions), where it will be possible to observe a meaningful signal from a base model. Regarding MGSM, small models only reach non-trivial performance with a relatively extensive token budget, and we kept training to 100B tokens to keep costs manageable as we experiment with many variables.
>
> > Relatedly, could you quantify the proportion of subwords that are (1) untrained and (2) undertrained after pretraining? This would help explain the mechanism behind the Universal Tokenizer's effectiveness.
>
> For models trained with the Universal tokenizer, we include the expanded languages at 5% to avoid undertrained tokens. However, to make sure that the presence of expanded language data is not to blame for the observed advantage of models trained with the Universal tokenizer, we ablate that percentage down to 0% in Section 5.4, and find that the advantage holds even when none of the expanded language data is present in pretraining.
>
> > Since the Universal Tokenizer covers languages unseen during pretraining, many subword embeddings might be undertrained or even untrained, especially for low-resource or distinct-script languages. Can the authors provide intuition for why these embeddings still facilitate adaptation in post-training stages?
>
> We hypothesize that the advantage of using the universal tokenizer has to do with better compression ratio in adaptation languages (see Appendix E for an extended discussion), leading to lower sequence lengths, and therefore more efficient representations. Compression ratio has been found to correlate with downstream performance (Goldman et al., 2024; Galle, 2019), although lower compression is not a sufficient condition for a better tokenizer (Schmidt et al., 2024). This, in combination with shared tokens across languages, may begin to explain the advantage of the universal tokenizer.
>
> > OFA [1] is another wise embedding initialization strategy for CVA; including this could further strengthen the completeness of the related work.
>
> We are thankful for the pointer to more advanced CVA methods. We will extend the CVA portion of the Related Work section to include these more modern methods. OFA, while an intriguing method, changes the parameter count, and in this experiment we seek to keep the parameter count the same in order to isolate variables. In fact, in the OFA paper, one of the baselines is to copy over shared embeddings and randomly initialize new tokens. Although comparison to advanced CVA techniques would certainly be informative, we would like the focus of this paper to remain on tokenizer design, as Reviewer **UXUT** has kindly recognized.

---

> > ### Comment · Reviewer_UXUT · 2025-11-25
> > **some doubts after reading the other reviews**
> >
> > While I appreciate the controlled experimental setup that
> > yields several valuable insights, some of the key findings
> > -- as noted by other reviewers as well -- are not
> > sufficiently novel, having already been discussed or
> > identified in prior work (e.g., better performance may come
> > from the lexical overlap between similar languages due to an
> > appropriate tokenizer). As a result, I now think that the
> > overall contribution of the paper is not as strong as I
> > initially thought. I have therefore lowered my score to 6.

---

> > > ### Author Response · Authors · 2025-12-03
> > > **Author Response**
> > >
> > > We thank Reviewer **UXUT** for their responsiveness, but regret to see the lowered score and appreciate the reviewer’s time and feedback.
> > > The novelty of our work is in our comprehensive pretraining experiments, which is commonly under-valued but is a necessity for such research. We show that having as many languages as possible in the tokenizer does not have a diminishing gain (even if they are extremely underrepresented during pretraining) but leads to much better adaptation. This is a significant result and very impactful for developing multilingual LLMs. We agree that shared tokens are almost certainly one of the reasons that the universal tokenizer results in a model with greater language plasticity, as explained in our previous comment. However, we also see improved adaptation on unseen languages, where there is less lexical overlap.
> > > We would have appreciated citations of prior work that report the same findings, so that we can better contextualize our work, had the rebuttal process not been frozen.

---

### Official Review · Reviewer_5iVL · 2025-11-01

**Soundness:** 2
**Presentation:** 3
**Contribution:** 2
**Rating:** 4
**Confidence:** 4

**Summary:**

The main contribution of this paper is to propose training the tokenizer on a large variety of language corpora before pretraining (on a pretraining dataset of lower diversity). The paper shows that such a tokenizer leads to negligible drops in performance on the pretraining languages, while yielding significant improvements on the expanded (seen-by-tokenizer) languages and smaller but notable gains on expanded (unseen-by-tokenizer) languages. The experiments are extensive and cover multiple languages and setups.

**Strengths:**

1. Addresses an important challenge in LLMs regarding the utility of tokenizers for diverse languages.
2. Experiments are extensive, and those sections are well written and clearly explained.
3. An improved multilingual tokenizer would be of broad interest to the multilingual LLM community.

**Weaknesses:**

1. The expanded (seen languages) setup, which is the main focus of the paper, feels somewhat artificial. Since the Universal tokenizer sees data from all languages, it seems like an unfair comparison because if pretraining data for these languages are already available to build the tokenizers, one could simply create a single tokenizer using an upweighting strategy and then pretrain on all available data. That would likely be the practical approach. It is difficult to imagine a real-world scenario where corpora from two languages are available, yet pretraining is only done on one of them (while the tokenizer is trained on both).
2. The paper would benefit from a deeper analysis of why the UNIVERSAL tokenizer helps during pretraining. Section 5.4 suggests benefits even when 0% of the language is included in pretraining, but it is unclear why a token for an expanded language would be useful for the Universal tokenizer unless it undergoes some “hits” during training. Otherwise, those token embeddings would remain close to their initialization. One hypothesis is that the observed improvements occur primarily for expanded languages sharing the same script, while another is that the filtering method still fails to fully remove contamination. A clearer analysis would strengthen the paper’s contribution and improve understanding of the benefits of the Universal tokenizer.
3. Overall, the method is slightly unclear. While the analyses are strong, the lack of sufficient discussion about the differences between UNIVERSAL and CLUSTER makes it difficult to understand where the gains are coming from.

I lean toward a reject in its current form, mainly because the seen-by-tokenizer setup which largely comprises the paper feels somewhat artificial based on my understanding.

**Questions:**

My questions below reflect what I found unclear.
1. I don’t fully understand what the CLUSTER tokenizer is and how it differs from UNIVERSAL. Is it that UNIVERSAL sees both expanded and cluster-specific languages for each of the three clusters, while CLUSTER does not?
2. **L127** *“Half of the training mix consists of an even distribution of all languages in the instruction finetuning data.”* This may be a nitpick, but I’m not sure this qualifies as “CPT,” given that it uses instruction tuning data. Could the high quality of this finetuning data (relative to typical pretraining data) be contributing to the tokenizer’s effectiveness? Additionally, why was an even distribution chosen?
3.  **L246** *“Where all languages are uniformly weighted except English”*: does this mean that weights are proportional to data availability?
4. Regarding the CLUSTER tokenizer: is this a unique tokenizer specific to each cluster (i.e., three separate tokenizers, each of size 250k)? If so, does Figure 2 represent three separate language models trained on these clusters, and then CPT without any tokenizer intervention on the expanded languages? This setup seems somewhat unintuitive, as one would expect a baseline such as CVA applied to the expanded languages on top of the cluster tokenizer in that figure.

---

> ### Author Response · Authors · 2025-11-21
> **Official Comment by Authors**
>
> We thank **Reviewer 5iVL** for highlighting the importance and relevance of the problem we’ve chosen to tackle, as well as the extensiveness and clarity of our experiments, and giving us the opportunity to clarify some points.
>
> > Since the Universal tokenizer sees data from all languages, it seems like an unfair comparison because if pretraining data for these languages are already available to build the tokenizers, one could simply create a single tokenizer using an upweighting strategy and then pretrain on all available data. That would likely be the practical approach. It is difficult to imagine a real-world scenario where corpora from two languages are available, yet pretraining is only done on one of them (while the tokenizer is trained on both).
>
> This is representative of the real-world scenario where “expanded” languages represent “new” languages that were not the focus of LLM-developers at the time of pretraining. Conducting additional training or fine-tuning to improve performance on these languages is extremely common- as an example, SeaLLM v1 is the result of CPT and SFT on LLaMA 2 for South-East Asian languages, of which some were included in the original pretraining (e.g. Chinese, Vietnamese) and other languages are unseen (e.g. Lao, Khmer). We propose having all languages in “at least” in the tokenizer mixture to enable better adaptation.
>
>
> > I don’t fully understand what the CLUSTER tokenizer is and how it differs from UNIVERSAL. Is it that UNIVERSAL sees both expanded and cluster-specific languages for each of the three clusters, while CLUSTER does not?
>
> This is correct - the CLUSTER tokenizer is only trained on the pretraining target languages for that cluster model, while the UNIVERSAL cluster is trained on the languages of all of the clusters. Therefore, there is only one UNIVERSAL tokenizer trained for all the cluster models.
>
> > L127 “Half of the training mix consists of an even distribution of all languages in the instruction finetuning data.” This may be a nitpick, but I’m not sure this qualifies as “CPT,” given that it uses instruction tuning data. Could the high quality of this finetuning data (relative to typical pretraining data) be contributing to the tokenizer’s effectiveness? Additionally, why was an even distribution chosen?
>
> Use of instruction data has become increasingly common in “mid-training (or cooldown)” of many LLMs such as SmolLM [1] and OLMo [2]. We chose the term CPT to reflect the broad data used, including English sources and all adaptation languages together, as well as to separate it from pretraining. Our SFT experiments, which are only done using the data from a particular adaptation language, isolate whether or not the observed effect is due to high quality finetuning data, and the advantage of the universal tokenizer holds in that setting as well (see section 4.2 for an extended discussion). We will make this point clear in the final version of the paper. An even distribution for language data in CPT was chosen because each language has roughly the same amount of data (15k samples).
>
> [1] “SmolLM2: When Smol Goes Big.” arXiv:2502.02737
> [2] “2 OLMo 2 Furious.” arXiv:2501.00656
>
> > L246 “Where all languages are uniformly weighted except English”: does this mean that weights are proportional to data availability?
>
> In creating the pretraining data mix, we regard English in a separate weighting category due to extensive data availability and the necessity for a large amount of English data for improved performance on other languages. This is quite common in many pretrained LLMs such as SmolLM, where English data varies from 90-58% across different stages of training [1]. Therefore, we set the weight for English pretraining data manually, and all other languages are uniformly weighted within the allocation for multilingual data.
>
> > Regarding the CLUSTER tokenizer: is this a unique tokenizer specific to each cluster (i.e., three separate tokenizers, each of size 250k)? If so, does Figure 2 represent three separate language models trained on these clusters, and then CPT without any tokenizer intervention on the expanded languages? This setup seems somewhat unintuitive, as one would expect a baseline such as CVA applied to the expanded languages on top of the cluster tokenizer in that figure.
>
> Yes, the cluster tokenizer is specific to each cluster- meaning, the European cluster tokenizer is trained only on languages in the European cluster, the Asian tokenizer on languages in the Asian cluster, etc. Figure 2 represents the results of CPT on 6 models – two for each cluster (one model trained with the universal tokenizer, one model trained with the cluster tokenizer), three clusters total. CVA is applied on the cluster tokenizer in that figure – so, in the CVA comparison experiment, we first expand the tokenizer vocabulary, instead of directly doing CPT using the same cluster tokenizer.

---

> > ### Comment · Reviewer_5iVL · 2025-11-26
> >
> > Thank you authors for their rebuttal and for addressing some of my earlier questions.
> >
> > Regarding the point about including all languages “at least” in the tokenizer mixture: I still find this experimental setup somewhat unconvincing. While I understand the motivation behind proposing a universal tokenizer that covers all languages, doing so generally requires access to at least some corpus for each of those languages in order to train the tokenizer. So for example, once developers have obtained such data for languages like Lao or Khmer, it seems unlikely that they would include these languages in the tokenizer but intentionally exclude them from the pretraining mixture. If SeaLLM v1 had access to Lao and Khmer I see no reason why they would decide to not train on it in addition to other languages and only build a tokenizer that incorporated Lao and Khmer.
> >
> > In real-world scenarios, even if only a small amount of data is available for a low-resource language, it is typically included within the training corpus. When more data becomes available later, developers can continue pretraining or fine-tuning to further improve performance in that language, much like the scenario the paper describes.
> > For this reason, I find the paper’s setup where a language is present in the tokenizer but entirely absent from the pretraining data to reflect an unrealistic deployment or development pipeline.
> >
> > This is my main concern with the setup.

---

> > > ### Author Response · Authors · 2025-12-03
> > > **Author Response**
> > >
> > > We thank Reviewer **5iVL** for their responsiveness, and the opportunity to discuss further.
> > >
> > > We completely agree that model creators should find corpora for as many languages as possible, and include them in the tokenizer training and pretraining, even if they don’t plan to officially support these languages. This is exactly our proposal, and corresponds to the Universal tokenizer setup, where this data is included in tokenizer training and model pretraining. However, this is not always the case, where they only train the tokenizer and model on the officially supported languages – this scenario maps to the Cluster tokenizer setup in our work. In section 5.4, we ablate the presence of expanded language data in pretraining in order to isolate language plasticity as originating from the tokenizer. This is the only time we include language data in the tokenizer training and not in pretraining – not as a realistic setup. We also test adaptation to unseen languages that aren’t included in either tokenizer training or pretraining, and find that the universal tokenizer adapts better to those languages as well.
> > >
> > > In the SeaLLM example, Lao and Khmer are unseen languages, and wouldn’t have been included in tokenizer training or model training. Chinese and Vietnamese, being present in pretraining and tokenizer training but are the target for posttraining, would correspond to expanded languages. To state our claim in terms of the SeaLLM example, we advise that including as many languages as possible in the tokenizer (including potential expanded languages like Chinese and Vietnamese), helps to set the model up for more efficient language adaptation in posttraining, even to unseen languages (like Lao and Khmer).
> > >
> > > This seems to be a simple misunderstanding about our setup, and how it is intended to reflect real-world problems – in the camera-ready paper, we will make sure this is clearly explained in the introduction.

---

### Official Review · Reviewer_WPps · 2025-11-01

**Soundness:** 2
**Presentation:** 3
**Contribution:** 3
**Rating:** 4
**Confidence:** 4

**Summary:**

This paper investigates how tokenizer design affects the multilingual adaptability of large language models. The authors propose using a universal multilingual tokenizer trained over a wide range of languages during pretraining, rather than adapting tokenizers post hoc. Through experiments spanning more than 60 languages and multiple adaptation settings, they find that models pretrained with a universal tokenizer achieve strong gains on low-resource and unseen languages, while maintaining performance on high-resource ones. The approach yields faster adaptation, improved translation and instruction-following quality, and outperforms cross-lingual vocabulary adaptation baselines. The study highlights tokenizer choice as a simple yet powerful lever for enhancing multilingual plasticity (Chen et al 2023) in language models.

**Strengths:**

- The paper presents a clear and thoughtful study of how tokenizer design influences multilingual adaptability.

- Its originality lies in approaching language plasticity (Chen et al 2023) through a simple, pretraining-time intervention rather than architectural or post-hoc changes.

- The experiments are broad, well controlled, and convincingly demonstrate that a universal tokenizer improves cross-lingual transfer without harming high-resource performance.

**Weaknesses:**

- The experiments, though broad, are confined to mid-sized models; it remains unclear whether the observed gains in adaptation speed and multilingual coverage hold at larger scales.

- The evaluation relies heavily on LLM-as-judge metrics. Adding human or cross-judge validation, especially for languages with distinct scripts, would bolster confidence in the reported improvements.

- The paper’s discussion of prior work on language plasticity is limited. It briefly cites Chen et al. (2023) to define the term but does not meaningfully compare its tokenizer-based approach with training-time interventions such as active forgetting or embedding resets. A fuller discussion would help clarify what is novel about influencing plasticity through tokenizer design rather than optimization dynamics.  Both works address the question of improving multilingual plasticity through pretraining-time interventions, but via very different mechanisms. So it would strengthen the related work section to discuss this connection more _explicitly_.

**Questions:**

1. Could the authors elaborate on how their tokenizer-based intervention differs conceptually from prior training-time approaches to improving language plasticity, such as active forgetting (Chen et al., 2023)?

2. Do the observed multilingual gains and faster adaptation trends hold at larger scales or in more compute-intensive settings?

3. How robust are the results to the choice of evaluation method? For instance, would human evaluation or alternative judges confirm the same improvements, especially for languages with distinct scripts or tokenization structures?

---

> ### Author Response · Authors · 2025-11-21
> **Official Comment by Authors**
>
> We would like to thank **Reviewer WPps** for their thoughtful feedback, and recognizing the originality of our approach, extensiveness of our experiments, and clarity in writing.
>
>  > Could the authors elaborate on how their tokenizer-based intervention differs conceptually from prior training-time approaches to improving language plasticity, such as active forgetting (Chen et al., 2023)?
>
> Our approach is unique in targeting the tokenizer for improved language plasticity capabilities. The novelty of our approach is that even for the underrepresented languages (in pretraining data), including them in tokenizer training enables language expansion after the post-training as opposed to taking care of this in training time. We believe that this is particularly important for large-scale LLM pretraining, where including more language is costly (due to capacity and language balancing). As opposed to our proposal, more sophisticated training-time approaches are not adapted in large-scale pretraining and require validation. We will add a comparison for these approaches in the introduction, and include an overview of other language plasticity methods in the related work section. We are grateful **Reviewer WPps** has identified this gap.
>
> > Do the observed multilingual gains and faster adaptation trends hold at larger scales or in more compute-intensive settings?
>
> Due to the extensive nature of our experiments and the cost of pretraining, we work with a mid-sized model (3B parameters) We believe this model size is sufficient to draw significant conclusions, and we anticipate they would also apply to larger models and token budgets, as supported by previous research (Biderman et al., 2023; Longpre et al., 2024; Aryabumi et al., 2024).
>
> > How robust are the results to the choice of evaluation method? For instance, would human evaluation or alternative judges confirm the same improvements, especially for languages with distinct scripts or tokenization structures?
>
> Due to the time and cost associated with human evals, we rely on LLM as a judge for evaluation. For additional validation beyond LLM-as-a-judge, we also report machine translation results in section 4.1, demonstrating the advantage of using a universal tokenizer across evaluation tasks. In the Flores-200 test set (XX→EN), our universal tokenizer enables an average of 5.3 BLEU increase (a relative 53% increase – 15.2 vs 9.9) for expanded languages and 2.2 BLEU increase in primary languages (14% relative improvement – 17.9 vs 15.7).

---

> ### Comment · Reviewer_WPps · 2025-11-28
>
> Thank you for the focused rebuttal. The promised revisions on the distinctions from training-time plasticity methods, the clarification of scaling expectations, and the added emphasis on MT results have addressed my concerns now. I will raise my score to 6, conditional on these promised revisions, once OpenReview allows me to do score updates.

---

> > ### Author Response · Authors · 2025-12-03
> > **Author Response**
> >
> > We would like to thank Reviewer **WPps** for their engagement and raising their scores. We commit to making these changes for the camera-ready paper.

---

### Author Response · Authors · 2025-12-03
**To Area Chairs**

Dear Area Chairs,

We appreciate the thoughtful feedback from our reviewers, and the opportunity to respond and provide clarification. We would like to summarize some of the main points raised during the rebuttal period here.

Reviewers WPps, 5iVL, UXUT note the importance of the problem we choose to tackle, WPps emphasizes the originality of our approach, reviewers 5iVL, UXUT recognize the clarity of the evidence we present, and our paper is recognized as thoughtful and well written by reviewers WPps, JwKn. Almost all reviewers highlight the extensiveness of our experiments.

Reviewers **WPps** and **UXUT** expressed some concern over the use of LLM-as-a-Judge as a metric in our paper. We selected this method due to the difficulty of evaluating open-ended generations. We emphasized machine translation as a metric as well, and reviewer **WPps** was planning to increase their score.

There was also some concern expressed by reviewers **WPps** and **UXUT** about the model size used in our experiments, and whether these findings would scale to a larger model size. Given the significant cost of pretraining and the extensiveness of our experiments, we selected a model size large enough (3B) to draw significant conclusions, but still reasonable given the constraints. We anticipate they would also apply to larger models and token budgets, as supported by previous research.

Reviewers **UXUT** and **JwKn** noted that the CVA embedding initialization methods we used as comparison are quite basic, and there have been more advanced methods presented recently. These works, such as FOCUS, often use mean or random initialization as a baseline, and we as well intend for the CVA methods we used to only be interpreted as a baseline. We do not claim that our proposed method outperforms CVA as a whole. We will underscore this point,  and bolster the CVA portion of our related work section to include an overview of these modern methods. We will also include a comparison to other training-time approaches for language plasticity, as noted by reviewer **WPps**.

Some reviewers, namely **5iVL**, had some confusion around our setup and how realistic it is. In our response, we clarified that we don’t expect that model practitioners would train the tokenizer on languages that aren’t included in pretraining. In our Universal tokenizer setup, “expanded” languages are languages that aren’t the focus of pretraining, and are included in the tokenizer training and pretraining at a small percentage. This is the approach that we recommend, as opposed to training the tokenizer only on primary pretraining focus languages, which is the standard approach. We also discussed a real-world example. We are happy to make this point more clear in the camera-ready paper.

Reviewer **JwKn** emphasized the reliance of our method on a relatively large vocabulary size, and claims significant cost in such a large vocabulary size. In our responses, we clarify that we keep the total number of parameters the same across vocabulary size experiments. We provide an overview of modern training methods, such as weight-tying, that reduce the cost of a larger vocabulary size to negligible. We also point to models such as Gemma 2B and 4B that use this vocabulary size, even at the same parameter scale, to illustrate the relevance of this approach.

Reviewers **JwKn** and **UXUT** also pointed out that the observed effect of improved language plasticity with a Universal tokenizer is likely due to shared tokens to minimize our contribution. We completely agree that this is likely one of the reasons, and will include an analysis of token overlap in the appendix. We show that having as many languages as possible in the tokenizer does not have a diminishing gain (even if they are extremely underrepresented during pretraining) but leads to much better adaptation, and we would like to emphasize that this is not a trivial result. Even though tokenizer work is undervalued, it’s one of the most crucial areas in LLM design, especially as pertaining to multilingual LLMs, and reducing the disparity in performance on low-resource languages.

Thank you for your time and attention in considering our work. We were hopeful that with these additional clarifications, especially around our setup and contributions, we would have come to a point of agreement with the reviewers. We are happy to make any additional changes that would strengthen our work.

---

### Meta-Review · Area_Chair_T8rD · 2025-12-19

**Summary:**

This paper investigates how tokenizer design influences the multilingual adaptability of large language models, specifically focusing on language plasticity. The authors propose a universal tokenizer trained on a broad set of languages before pretraining to facilitate more efficient post-training adaptation. Extensive experiments across over sixty languages demonstrate that this simple intervention significantly improves performance on low-resource and unseen languages while maintaining high-resource performance

**Strengths:**
1. The research provides a clear and well-executed study of language plasticity through a simple low-cost pretraining-time intervention rather than complex architectural changes.
2. The experiments are extensive and controlled, demonstrating that a universal tokenizer enhances cross-lingual transfer across dozens of diverse languages without harming high-resource language performance.
3. It outperforms cross-lingual vocabulary adaptation baselines and shows faster adaptation and better compression efficiency, adding practical value to its theoretical contributions.

**Weaknesses:**
1. The primary experimental setup involving seen-by-tokenizer languages is criticized as somewhat artificial and less fair since pretraining data for those languages could have been included if it was available for tokenization.
2. The study focuses exclusively on mid-sized models of 3.3B parameters. It's unclear if gains hold at larger scales, as multilingual capability often scales with model size.
3. The evaluation relies heavily on LLM-as-judge metrics, which could be strengthened by human validation or alternative judges to confirm results across distinct scripts.
4. The comparison with baseline adaptation methods is limited to simple random or mean initialization rather than more recent state-of-the-art cross-lingual vocabulary adaptation techniques. Stronger, recent methods would provide a fairer reference, though the focus is on tokenizer design.

**Reviewer Concerns:**

The paper received an average rating of 4.00, with a large variance (Min: 0, Max: 8), indicating significant disagreement among reviewers. After the authors submitted their rebuttal, reviewers updated their views:
Reviewer WPps stated that the promised revisions, including clarifying distinctions from training-time plasticity methods, explaining scaling expectations, and emphasizing machine translation results, have addressed his/her concerns, and he/she will raise the score to 6.
Reviewer 5iVL remains unconvinced by the experimental setup, particularly the scenario where a language is included in the tokenizer but entirely absent from pretraining data, viewing it as an unrealistic deployment or development pipeline.
Reviewer UXUT, who initially gave the highest rating, now considers the paper's overall contribution less strong after reading other reviews and will lower the score to 6.
Reviewer JwKn, the most critical, believes the rebuttal did not adequately address his/her concerns, so he/she did not upgrade their rating. He/She notes the paper fails to quantify the effect of a larger tokenizer or show that vocabulary adaptation is insufficient to address the problem with minimal compute cost.

**Reviewer Scores:**

Considering that Reviewer WPps may increase the score, Reviewer 5iVL may decrease his/hers, and the other two reviewers are likely to keep their scores unchanged, the average score of the paper may remain the same, lacking strong support.

---

### Decision · Program_Chairs · 2026-01-26

Reject